## [Peer Review File · Biology Open]

GelMA hydrogel stiffness influences epithelial to mesenchymal transition in MCF7 but not MDA-MB-231 breast cancer cells in 3D culture

Jessika A. Wise, Margaret J. Currie, Tim B. F. Woodfield, Khoon S. Lim and Elisabeth Phillips

DOI: 10.1242/bio.062212

Editor: Christopher A. Maher

Review timeline

Original submission:	5 November 2025
Editorial decision:	14 November 2025
First revision received:	12 January 2026
Accepted:	20 January 2026

Original submission

First decision letter

MS ID#: bio.062212

MS Title: GelMA hydrogel stiffness influences epithelial to mesenchymal transition in MCF7 but not MDA-MB-231 breast cancer cells in 3D culture

Authors: Jessika A Wise; Margaret J Currie; Tim B F Woodfield; Khoon S Lim; Elisabeth Phillips

I have now reached a decision on the above manuscript.

The reviewer reports are shown at the bottom of this email.

As you will see, the reviewers gave favourable reports, but raised some critical points that will require amendments to your manuscript. I hope that you will be able to carry these out, because we would like to be able to accept your paper.

At this stage, we also ask you to ensure your manuscript complies with our formatting guidelines - please see our manuscript preparation guidelines for details. Provided you are able to fully address the referees' comments, we are positive about publication of your paper (we accept over 95% of revision submissions) and therefore hope you won't mind any extra work involved in reformatting your manuscript at this point.

Please upload both a 'clean' version of your Word file, along with a highlighted version clearly showing where you have made changes in the revised manuscript. Please avoid using 'Track changes' in Word files as these are lost in PDF conversion.

I should be grateful if you would also provide a point-by-point response detailing how you have dealt with the points raised by the reviewers in the 'Response to Reviewers' box. Please attend to all of the reviewers' comments. If you do not agree with any of their criticisms or suggestions please explain clearly why this is so.

Reviewer 1

Comments for the author

Review of "Tuneable hydrogel stiffness in a 3D in vitro model induces epithelial to mesenchymal transition in MCF7 but not MDA-MB-231 breast cancer cells."

Wise and colleagues describe the use of GelMA hydrogels to analyze the effect of differential ECM stiffness on the commonly-used breast cancer cell lines, MCF7 and MDA-MB-231.

The Authors began by measuring key hydrogel features (mass swelling ratio, soluble fraction, etc.) across a range of experimental conditions (macromer concentration and/or number of tumor cells), demonstrating how ECM stiffness and spatial oxygen concentration can be controlled by varying the GelMA macromer concentration. These data will be useful for others intending to reproduce/build on this study. Next, the authors characterized tumor cell phenotypes over time in different experimental conditions, illustrating that increased macromer concentration (i.e., stiffness + lower oxygen), correlated with changes in key cellular functions (e.g., metabolic activity, proliferation, etc.). These data also revealed hints about the differences between MCF7 and MDA-MB-231 behaviors when cultured on GelMA hydrogels (Fig. 4), although the authors did not comment on these differences.

Next, the authors imaged stiff and soft GelMA hydrogels seeded with MDA-MB-231 and MCF7 cells and observed distinct growth patterns of each cell line in the core/center and different stiffnesses that were not entirely due to changes in proliferation (measured with KI67). Further, the authors measured the abundances of known markers of EMT (E-cadherin, N-cadherin and Vimentin) in both cell types and two hydrogel stiffnesses, revealing that MCF7 cells uniquely induce genes associated with EMT in stiff hydrogels. The authors note that these observations are compatible with previous work demonstrating altered EMT marker expression by MCF7 cells grown in collagen gels, and note that their work adds new context to these observations by highlighting the role of hydrogel stiffness in the phenomenon.

Overall, I find that the paper is very well written, making it easy for the reader to follow along with the Authors' logic throughout. Further, I believe that the differential responses of the MDA-MB-231 and MCF7 cell lines to distinct hydrogel conditions is well reasoned and that the experiments and analyses adequately support the majority of the claims. I believe that this manuscript should be published after addressing the following comments:

1. When discussing the observed stiff-MCF7-specific changes in ECM markers, the authors state that this suggests that "...increased stiffness of ECM in hormone receptor positive (Luminal A) breast cancers may promote breast cancer cell invasiveness via EMT." Indeed, changes in ECM marker expression is not observed in the triple-negative MDA-MB-231 cell line, but the authors note that this is likely because these cells have undergone partial EMT - not necessarily because it is a triple negative cell line. As a result, the claim that luminal A breast cancer invasiveness is regulated by stiffness not logically consistent.
2. Figure 5-7 images are blurry - should replace with high-quality images.

Reviewer 2

Comments for the author

Peer Review: "Tuneable hydrogel stiffness in a 3D in vitro model induces epithelial to mesenchymal transition in MCF7 but not MDA-MB-231 breast cancer cells"

This article describes the use of tuneable hydrogels as a 3D in-vitro culture method to study the response of immortalized human breast cancer cells such as MCF7 and MDA-MB-231. The article

main aim is to design 3D models mimicking the evolving breast tumour microenvironment (TME). The authors compare two distinct breast cancer cell lines to study the breast cancer subtype-specific responses to matrix stiffness. By employing clinically relevant hydrogel stiffnesses (GelMA at 5%, 7.5%, 10% wt%) that span the range from normal breast tissue (~10 kPa) to high-grade tumors (~30–40 kPa), ensuring the *in vitro* model reflects *in vivo* conditions. Importantly, the team confirmed the actual compressive moduli of these hydrogels (≈ 11 –33 kPa with cells) and even measured oxygen gradients across the hydrogel depth. This thorough characterization strengthens confidence that observed cellular changes are indeed due to stiffness differences rather than unintended culture artifacts. The suite of assays used is comprehensive – assessing cell viability, metabolic activity, proliferation, 3D growth morphology, and EMT marker expression – providing a multi-angle evaluation of cell behavior. The inclusion of spatial analysis (center vs. edge) is an innovative aspect that accounts for gradient effects (nutrient/O₂ diffusion), adding depth to the experimental quality. Overall, the experiments are well-controlled and generally support the conclusions that stiff matrices can induce EMT-related changes in MCF7 cells but not in MDA-MB-231 cells.

Despite these strengths, a few **specific issues** in experimental execution and presentation should be addressed to ensure maximal rigor and clarity:

1. Article structure:

- The first Results section, spanning from hydrogel characterization through proliferation and growth behavior (up to Figure 6), is dense and difficult to parse. The inclusion of diverse metrics—mechanical data, cell viability, morphology, proliferation, and spatial distribution—without sub-sectioning may overwhelm readers. Structuring the Results section into clearly titled subsections (e.g., “Hydrogel Mechanical Characterization,” “Viability and Proliferation in 3D Culture,” “Spatial Growth Patterns”) would improve readability and help contextualize the logic of the experimental progression.

2. Important definition of concepts:

1. **Inconsistency Regarding Phenotypic Maintenance in Soft Hydrogels**
Page 2, lines 29–30: The authors claim that “soft hydrogels maintained the phenotype of both cell lines,” which appears to contradict a statement earlier in the abstract asserting that phenotype changes occur in response to hydrogel culture. If encapsulation in GelMA matrices leads to phenotypic shifts, even in soft conditions, this should be clearly delineated. Conversely, if soft hydrogels are intended as a phenotypic baseline, that needs to be more rigorously supported and stated with consistency throughout the manuscript. The current phrasing introduces conceptual ambiguity about whether soft hydrogels represent a truly inert control or also induce microenvironmental changes.
2. **The use of “Extracellular Matrix- ECM” for GelMA:** *Page 4, line 14:* The authors refer to investigating the effects of “ECM stiffness” using GelMA hydrogels. While GelMA is a gelatin derivative derived from collagen, it is an engineered biomaterial and not a native

extracellular matrix per se. Although it serves as a proxy for ECM-like mechanical properties, the term “ECM” should be used with care, especially in mechanistic contexts. It would be more precise to describe GelMA as a “biomimetic ECM model” or “synthetic hydrogel system with tunable stiffness,” rather than as ECM itself. I would advise the authors to wrap this concept around the hydrogel system rather than the ECM because they are very distant from each other. Other commercial products like Matrigel resemble more closely the ECM conditions.

3. **Caution Regarding Claims About Tumor Microenvironment (TME) Modeling:** *Page 4, line 20:* Similarly to what’s stated right above this paragraph, authors state that “this study provides new insights into how breast cancer cells behave in response to ECM stiffness within the breast TME.” However, GelMA hydrogels, while valuable, lack the biochemical complexity and heterogeneity of the in vivo ECM. Native ECM contains a diverse set of components—glycoproteins, proteoglycans, and fibrous proteins—that dynamically evolve during tumor progression. Moreover, tumor-derived ECM is cell-generated and continuously remodeled, introducing spatiotemporal gradients and architectural complexity that are not replicated in static GelMA constructs. While the study offers meaningful data on stiffness-driven effects in a controlled environment, extrapolating to the TME should be qualified as modeling *aspects of* the ECM rather than the TME in full.

3. Experimental design and results:

1. **Intermediate Stiffness Condition (7.5% GelMA):** The authors decided to exclude the 7.5% (“medium”) stiffness condition from later analyses, stating that it did not show significant differences from soft or stiff hydrogels. While this simplification is understandable, it risks overlooking subtle trends. For completeness, key experiments (e.g., EMT marker expression) should ideally include the intermediate stiffness. This would confirm whether there is a threshold stiffness required to induce EMT or a more gradual transition. If data were collected but not shown, the authors could briefly report that no notable EMT changes were observed at 7.5% to justify its exclusion. Otherwise, the omission should be acknowledged as a limitation, since an intermediate point could strengthen the conclusion about stiffness-dependent effects.
2. **Replicates and Data Consistency:** The manuscript would benefit from clarifying the number of biological replicates for each experiment and ensuring consistency between groups. For instance, in Figure 4, the MDA-MB-231 viability assay had $n=4$ replicates whereas MCF7 viability had $n=3$, and the metabolic activity assay used $n=5$ for MDA-MB-231 vs. $n=3$ for MCF7. This imbalance is not explained. All else being equal, both cell lines’ assays should use the same n , so any discrepancy should be justified (e.g., if a replicate was lost due to technical issues). Similarly, it’s implied that separate sets of hydrogels were used for viability, metabolism, and proliferation assays at each time point – the authors should confirm this in the methods. Clearly defining what constitutes a replicate (individual hydrogel, independent experiment, etc.) would help readers judge the robustness of the data.

3. **Cell Density Effects on Hydrogel Properties:** The authors wisely tested whether increasing the cell encapsulation density alters the hydrogel's mechanical properties. They report that adding cells up to 5×10^6 /mL did not significantly change stiffness in 5% and 7.5% gels, though a modest reduction was seen at 10% GelMA. Based on these results, 5×10^6 cells/mL was chosen for experiments. This is a sound choice balancing cell viability and matrix integrity. However, the description of the cell-density experiment is a bit confusing. The text states that adding *either* cell type at 5×10^6 /mL “reduced the stiffness” of 10% gels, but “the effect was not concentration-dependent”. It's unclear whether higher cell concentrations (10 or 15 million/mL) caused similar or greater softening. The authors should clarify this point. If 5×10^6 was the lowest density that caused a significant stiffness drop (and higher densities did not further reduce stiffness), that finding should be explicitly stated. Currently, the phrasing could be read as an inconsistency (why pick a cell density that already measurably softens the matrix?). A brief clarification will resolve any confusion and confirm that the chosen cell density does not confound the stiffness conditions beyond acceptable limits.
4. **Controls and Baselines:** Experimental controls are mostly appropriate. The use of cell-free hydrogels as controls for measuring swelling, soluble fraction, and stiffness is excellent, as it allows the authors to distinguish the influence of the encapsulated cells on these properties. For viability and metabolic assays, “negative controls” were mentioned (presumably acellular hydrogels or media-only background) with results normalized accordingly – this is good practice to account for background fluorescence. One control that might be lacking is a 2D culture comparison or an initial phenotype baseline for the cells. While the focus is on 3D stiffness effects, having a reference point (e.g., cells cultured on plastic or in very low-density 3D matrix) could contextualize how much 3D culture alone induces changes. Are authors using Day 1 in 3D as the reference state? If so, this should be clarified (maybe it was but I failed to understand it). Additionally, immunostaining controls (such as isotype or secondary-only staining) are not described; presumably they were done, but mentioning them would assure readers that the fluorescence quantification is specific.
5. **Phenotypic and Morphological Assessment:** The evidence for MCF7 cells undergoing EMT in stiff hydrogels relies on marker expression changes and some altered growth patterns, but direct morphological signs of EMT (e.g., loss of cell-cell adhesion, adoption of spindle-like shape) were not clearly reported. The images in Figure 5 and Figure 7 should be scrutinized for qualitative changes. MCF7 cells generally grew as clusters in both soft and stiff matrices (as shown in Fig.5c, day 21 images), indicating they did not fully disperse or invade the matrix as single cells. Even in stiff conditions, MCF7 still formed aggregates, albeit with different size/coverage at the edge vs. center. This suggests a partial EMT or mixed phenotype. The authors should comment on any observable changes in cell morphology or colony organization with stiffness. For example, did MCF7 cells in 10% GelMA exhibit more protrusive or mesenchymal shapes at the single-cell level, or were they simply forming larger spheroids? This is not a flaw per se, but discussing it would provide a more nuanced picture of the EMT state
 - In line with this the study's conclusion that stiff matrices induce EMT in MCF7 cells is primarily based on a decrease in E-cadherin and an increase in N-cadherin and vimentin. While these markers are indeed canonical indicators of EMT, the process is

not binary, and cancer cells can exist in hybrid epithelial/mesenchymal states. To more conclusively demonstrate EMT, additional markers and functional assays would be beneficial:

- Epithelial markers: Assessing additional markers such as EpCAM or cytokeratins could provide deeper insight into epithelial status.
- EMT transcription factors: Expression of Snail, Slug, or Twist (via immunostaining or qPCR) would help confirm that the EMT program is transcriptionally engaged.
- Signaling pathways: Examination of upstream signaling nodes such as TGF- β , YAP/TAZ, or β -catenin could illuminate the mechanistic basis of EMT induction by stiffness.
- Functional assays: Assessing motility or invasiveness (e.g., spheroid spreading assays, invasion into matrix) would directly link molecular changes to functional outcomes.

Without these additional data, the claim of stiffness-induced EMT should be **interpreted as suggestive but not definitive**. Acknowledging this limitation would improve the scientific balance of the discussion.

6. **Cell Viability in Stiff Hydrogels:** An important experimental observation is that cell viability decreased in the stiffer matrices, especially for MDA-MB-231. By day 21, MDA cells had significantly lower viability in 7.5% and 10% gels (compared to soft 5%) starting from day 7 onward. MCF7 viability remained high in 7.5% and only dropped in 10% gels at the final day. This indicates that the stiffer environment imposed greater stress on cells (particularly the already-invasive MDA line). The authors should ensure this point is discussed in terms of experimental quality: does the reduced viability in stiff conditions affect the interpretation of EMT marker results? For instance, if a fraction of MDA-MB-231 cells died under stiffness stress, the surviving population might be the hardier (potentially more mesenchymal) cells – yet no EMT marker change was seen, reinforcing that their phenotype was stable.
7. **Image Sample Size:** When quantifying images for Fig.5–7, how many fields or sections were analyzed per hydrogel? The methods indicate that 30 μ m cryosections were taken and stained, but it doesn't specify if, say, three sections per hydrogel were imaged, or multiple regions per section. The $n=3$ for imaging data suggests 3 hydrogels, but likely multiple images from each were averaged. Stating something like “for each hydrogel, 3–5 fields in both center and edge regions were quantified and averaged” would be useful for others aiming to reproduce the imaging analysis with adequate sampling. If only one section/field

per region per hydrogel was used, that could be a source of variability – but presumably the authors took more.

4. Limitations of the study:

The Discussion does a good job of acknowledging certain limitations and alternative explanations. However, one limitation that could be more clearly stated is the translational relevance: while the model mimics tumor stiffness ranges, it is still a simplified system (single cell type in a uniform matrix). In reality, cells in tumors experience heterogenous stiffness, interact with stromal cells, etc. The authors might add a sentence noting that their 3D model is a reductionist system focusing on stiffness in isolation – which is a strength for mechanistic insight, but future work could incorporate additional tumor microenvironment factors.

In conclusion, the experimental approach is robust and well thought out. Addressing the points above, especially regarding the intermediate stiffness condition, replicate consistency, and clearer commentary on morphological versus molecular changes – will further solidify the experimental rigor. Overall, the **quality of experiments is high** and authors do not excessively user overclaims.

Reviewer 3

Comments for the author

Review: Tuneable hydrogel stiffness in a 3D in vitro model induces epithelial to mesenchymal transition in MCF7 but not MDA-MB-231 breast cancer cells

Summary: The authors present work which aims to elucidate the role of extracellular microenvironment mechanics with a particular focus on the TME stiffness and the use of hydrogels to study cancer cell dynamics and phenotype. The authors propose the use of a photoactivatable hydrogel which can be utilised to study the long-term culture of breast cancer cell lines. Overall, the manuscript is well written and, in most cases, very clear and straight forward. The experimental approach is well suited to the question the authors address and the results are well presented and robust. There are some slight deficiencies in the manuscript which if added would greatly improve the overall manuscript.

Strengths: The authors present a comprehensive temporal analysis of the effect of the hydrogel culture conditions on the EMT phenotype of two different breast cancer cell lines. In particular the temporal analysis adds confidence and robustness to the phenotypes and phenotypic switching that is observed in the MCF7 cell line.

Weaknesses: The study lacks orthogonal analysis of EMT switching. If the authors were able to show mRNA or protein expression levels are regulated via the stiffer hydrogel conditions with the MCF7 cell line, the results would be much more convincing if the authors utilised qPCR or western blot analysis to show these changes in Ncad or Ecad levels. It may be technically challenging to extract mRNA or protein samples from the hydrogels and this should be highlighted as a potential drawback of the technology and mentioned in the discussion, if so. The study also lacks an informative and comprehensive discussion would could be greatly improved by the following additions.

The authors could further elaborate in the discussion on a potential mechanism by which EMT is regulated by altered stiffness' of the hydrogel. One potential player could be YAP signalling which

responds to altered mechanical context and and plays a role in phenotypic switches in altered mechanical environments.

One potential application of the technology would be to culture patient derived tumour cell lines in the hydrogel conditions to determine the potential of EMT phenotypic switching. Adding this to the discussion would highlight the potential that the technology has for personalised medicine.

Minor concerns

Immunofluorescent labels in all figures are sometimes difficult to visualise and would benefit from increased size and a different coloured background or moved from within the image to a label below the image.

Dapi staining in the images would benefit from being in a different colour as the dark blue on black background can make it difficult to visualise individual nuclei.

Reviewer's Responses to Questions

1. Experimental quality

a. Does each figure have the proper controls? Experiments have appropriate controls.
 b. Are experiments performed using appropriate methods that will answer the question (or test the hypothesis or support the observations) posed by the authors? Is the right tool used for the job? The authors test the appropriateness of the GelMA hydrogels on the longer term culture of two breast cancer cell lines and analysis EMT status, proliferation and metabolic rates over a 21 day period. The authors make use of two cancer cell lines of different molecular subtype and use a variety of methods to analyse cancer cell characteristics. The study would benefit from orthogonal analysis of EMT status as mentioned in the major concerns section, however, this may be technically difficult and if so, should be expanded on in the discussion.

c. Were the data analysed using appropriate statistical tests? Yes

2. Reproducibility

a. Were experiments in each figure performed using adequate number of biological replicates? Yes

b. Is there sufficient raw data to assess the rigor of the analysis? Yes

c. Does the methods section provide sufficient detail to permit reproducibility? Yes

3. Completeness

a. Are the author's conclusions supported by the data? Conclusions could be expanded as mentioned, in particular the discussion could elaborate on potential molecular mechanisms underpinning the differential responses of the two cancer cell lines. Further discussion on the EMT phenotypic switch and supporting literature on the role of stiffness driving cancer cell EMT switching would strengthen the manuscript.

b. Are there any flaws in the experimental design that invalidate the approach taken by the authors? No

c. Are there experiments that have not been performed, but if true would disprove the conclusion? If yes, and if such experiments would be costly or time-consuming to perform, do the authors acknowledge this in a discussion of the limitations? There are standard orthogonal methods that the authors could use to improve the confidence in their findings. Additional experiments elucidating the molecular mechanisms behind the EMT switch could be conducted to strengthen the manuscript, however it could be argued that these experiments are outside the scope of the study or technically challenging. For instance, the authors could investigate whether TGF β driven EMT is accelerated by culturing in the stiffer hydrogels. The authors could also investigate whether the

EMT switch is reliant on key mechanosensing pathways, such as integrin/cytoskeletal pathways or YAP mechanosensitive pathways.

4. Scholarship

- a. Do the authors cite and discuss the merits of relevant data that would argue against their conclusion? No, this must be strengthened in the discussion by addressing the drawbacks of the hydrogel culture system.
- b. Do the authors cite and discuss the merits of relevant data that would support their conclusion? Partially, as already mentioned, the authors should expand the discussion and speculate on the molecular mechanism behind their findings, as one of the key weak points of the manuscript is the lack of mechanistic explanation behind the findings. This could be mitigated by an elaboration in the discussion on potential alignment of the studies findings with existing literature.
- c. For techniques/methods manuscripts, Do the authors cite and discuss the current state of the field and clearly explain how the method improves the field? Yes, the authors clearly outline existing work utilising similar or the same hydrogel systems in the study of breast cancer.

Title: The authors use the term tuneable, which suggests that the stiffness of the hydrogels can be adjusted or altered dynamically. Rather, a range of stiffnesses of the hydrogels is established in the study by adjusting the amount of GelMA added and so I am not sure the term tuneable is appropriate.

Reviewer's Responses to Questions

Experimental quality

Does each figure have the proper controls?

If 'No', please indicate reasons in Comments for Author box below.

Reviewer #1:

- Yes

Reviewer #2:

- No

Reviewer #3:

- Yes

Were the data analyzed using appropriate statistical tests?

If 'No', please indicate reasons in Comments for Author box below.

Reviewer #1:

- Yes

Reviewer #2:

- No

Reviewer #3:

- Yes

Reproducibility

Were experiments performed using adequate number of biological replicates?

If 'No', please indicate reasons in Comments for Author box below.

Reviewer #1:

- Yes

Reviewer #2:

- No

Reviewer #3:

- Yes

Does the methods section provide sufficient detail to permit reproducibility?
If 'No', please indicate reasons in Comments for Author box below.

Reviewer #1:

- Yes

Reviewer #2:

- Yes

Reviewer #3:

- Yes

Completeness

Are the manuscript's conclusions supported by the data?
If 'No', please indicate reasons in Comments for Author box below.

Reviewer #1:

- Yes

Reviewer #2:

- Yes

Reviewer #3:

- No

Scholarship

Do the authors cite and discuss the merits of data that would argue for and against their conclusion?

If 'No', please indicate reasons in Comments for Author box below.

Reviewer #1:

- Yes

Reviewer #2:

- Yes

Reviewer #3:

- No

Does the manuscript title & abstract accurately reflect the contents of the manuscript, without hyperbole?

If 'No', please indicate reasons in Comments for Author box below.

Reviewer #1:

- Yes

Reviewer #2:

- Yes

Reviewer #3:

- No

First revision

Author response to reviewers' comments

Reviewer 1: Reivew of "Tuneable hydrogel stiffness in a 3D in vitro model induces epithelial to mesenchymal transition in MCF7 but not MDA-MB-231 breast cancer cells."

Wise and colleagues describe the use of GelMA hydrogels to analyze the effect of differential ECM stiffness on the commonly-used breast cancer cell lines, MCF7 and MDA-MB-231.

The Authors began by measuring key hydrogel features (mass swelling ratio, soluble fraction, etc.) across a range of experimental conditions (macromer concentration and/or number of tumor cells), demonstrating how ECM stiffness and spatial oxygen concentration can be controlled by varying the GelMA macromer concentration. These data will be useful for others intending to reproduce/build on this study. Next, the authors characterized tumor cell phenotypes over time in different experimental conditions, illustrating that increased macromer concentration (i.e., stiffness + lower oxygen), correlated with changes in key cellular functions (e.g., metabolic activity, proliferation, etc.). These data also revealed hints about the differences between MCF7 and MDA-MB-231 behaviors when cultured on GelMA hydrogels (Fig. 4), although the authors did not comment on these differences.

Next, the authors imaged stiff and soft GelMA hydrogels seeded with MDA-MB-231 and MCF7 cells and observed distinct growth patterns of each cell line in the core/center and different stiffnesses that were not entirely due to changes in proliferation (measured with KI67). Further, the authors measured the abundances of known markers of EMT (E-cadherin, N-cadherin and Vimentin) in both cell types and two hydrogel stiffnesses, revealing that MCF7 cells uniquely induce genes associated with EMT in stiff hydrogels. The authors note that these observations are compatible with previous work demonstrating altered EMT marker expression by MCF7 cells grown in collagen gels, and note that their work adds new context to these observations by highlighting the role of hydrogel stiffness in the phenomenon.

Overall, I find that the paper is very well written, making it easy for the reader to follow along with the Authors' logic throughout. Further, I believe that the differential responses of the MDA-MB-231 and MCF7 cell lines to distinct hydrogel conditions is well reasoned and that the experiments and analyses adequately support the majority of the claims. I believe that this manuscript should be published after addressing the following comments:

1. When discussing the observed stiff-MCF7-specific changes in ECM markers, the authors state that this suggests that "...increased stiffness of ECM in hormone receptor positive (Luminal A) breast cancers may promote breast cancer cell invasiveness via EMT." Indeed, changes in ECM marker expression is not observed in the triple-negative MDA-MB-231 cell line, but the authors note that this is likely because these cells have undergone partial EMT - not necessarily because it is a triple negative cell line. As a result, the claim that luminal A breast cancer invasiveness is regulated by stiffness not logically consistent.

We thank the reviewer for highlighting this point. We agree that the original wording overstated the relationship between breast cancer subtype and stiffness-driven responses. While stiffness-associated EMT marker changes were observed in MCF7 cells and not in MDA-

MB-231 cells, this difference likely reflects the baseline phenotypic state of the cell lines rather than hormone receptor status per se. MDA-MB-231 cells already exhibit mesenchymal-like features, which may limit further EMT-associated changes in response to increased matrix stiffness.

To address this, we have revised the text to remove subtype-specific claims and now frame our findings in terms of baseline epithelial versus mesenchymal phenotypes.

Page 9, line 255: "Increased matrix stiffness may promote EMT-associated phenotypic changes in breast cancer cells with an epithelial baseline phenotype, such as MCF7."

2. Figure 5-7 images are blurry - should replace with high-quality images.

We thank the reviewer for raising this point. The apparent blurriness of the images in Figures 5-7 is due to image compression during PDF conversion for submission. The original images are high-resolution and were used for all quantitative analyses.

We have ensured that high-quality, full-resolution images are included in the resubmitted version to avoid any loss of clarity.

Reviewer 2: Please note: The detailed reviewer comments were pasted into this author text-box. An identical version is also uploaded as a Word document for formatting clarity only—no additional content is included.

This article describes the use of tuneable hydrogels as a 3D in-vitro culture method to study the response of immortalized human breast cancer cells such as MCF7 and MDA-MB-231. The article main aim is to design 3D models mimicking the evolving breast tumour microenvironment (TME). The authors compare two distinct breast cancer cell lines to study the breast cancer subtype-specific responses to matrix stiffness. By employing clinically relevant hydrogel stiffnesses (GelMA at 5%, 7.5%, 10% wt%) that span the range from normal breast tissue (~10 kPa) to high-grade tumors (~30-40 kPa), ensuring the in vitro model reflects in vivo conditions. Importantly, the team confirmed the actual compressive moduli of these hydrogels (\approx 11-33 kPa with cells) and even measured oxygen gradients across the hydrogel depth. This thorough characterization strengthens confidence that observed cellular changes are indeed due to stiffness differences rather than unintended culture artifacts. The suite of assays used is comprehensive - assessing cell viability, metabolic activity, proliferation, 3D growth morphology, and EMT marker expression - providing a multi-angle evaluation of cell behavior. The inclusion of spatial analysis (center vs. edge) is an innovative aspect that accounts for gradient effects (nutrient/O₂ diffusion), adding depth to the experimental quality. Overall, the experiments are well-controlled and generally support the conclusions that stiff matrices can induce EMT-related changes in MCF7 cells but not in MDA-MB-231 cells.

Despite these strengths, a few specific issues in experimental execution and presentation should be addressed to ensure maximal rigor and clarity:

(1) Article structure:

The first Results section, spanning from hydrogel characterization through proliferation and growth behavior (up to Figure 6), is dense and difficult to parse. The inclusion of diverse metrics—mechanical data, cell viability, morphology, proliferation, and spatial distribution—without sub-sectioning may overwhelm readers. Structuring the Results section into clearly titled subsections (e.g., "Hydrogel Mechanical Characterization," "Viability and Proliferation in 3D Culture," "Spatial Growth Patterns") would improve readability and help contextualize the logic of the experimental progression.

We thank the reviewer for this constructive suggestion. We agree that the first results section contains multiple experimental readouts and that clearer sub-sectioning would improve readability and guide the reader through the experimental logic.

To address this, we have restructured those results section into clearly titled subsections that reflect the progression.

- Hydrogel characterisation and cell incorporation (page 5, line 98)
- Cell viability, metabolic activity, and proliferation in 3D GelMA hydrogels (page 7, line 177)
- Cell morphology and spatial growth patterns in 3D GelMA hydrogels (page 7, line 190)

(2) Important definition of concepts:

» Inconsistency Regarding Phenotypic Maintenance in Soft Hydrogels

Page 2, lines 29-30: The authors claim that "soft hydrogels maintained the phenotype of both cell lines," which appears to contradict a statement earlier in the abstract asserting that phenotype changes occur in response to hydrogel culture. If encapsulation in GelMA matrices leads to phenotypic shifts, even in soft conditions, this should be clearly delineated. Conversely, if soft hydrogels are intended as a phenotypic baseline, that needs to be more rigorously supported and stated with consistency throughout the manuscript. The current phrasing introduces conceptual ambiguity about whether soft hydrogels represent a truly inert control or also induce microenvironmental changes.

We thank the reviewer for pointing this out. We have clarified that soft hydrogels (5wt%) maintained a stable phenotype from day 1 to day 21, based on EMT marker expression, whereas stiff hydrogels (10wt%) induced changes in EMT markers and morphology. This avoids suggesting that soft hydrogels are a 'baseline' while clearly showing the stiffness-dependent effects within the 3D culture system. We have rewritten that section of the abstract to clarify that.

Page 2, line 39: "Over a 21-day culture period, MCF7 cells exhibited partial epithelial-mesenchymal transition in stiff hydrogels, showing altered morphology, downregulating E-cadherin and upregulating N-cadherin and Vimentin. Comparatively, MDA-MB-231 cells showed no such changes. Phenotype remained stable in soft hydrogels for both cell lines."

» The use of "Extracellular Matrix- ECM" for GelMA: Page 4, line 14:

The authors refer to investigating the effects of "ECM stiffness" using GelMA hydrogels. While GelMA is a gelatin derivative derived from collagen, it is an engineered biomaterial and not a native extracellular matrix per se. Although it serves as a proxy for ECM-like mechanical properties, the term "ECM" should be used with care, especially in mechanistic contexts. It would be more precise to describe GelMA as a "biomimetic ECM model" or "synthetic hydrogel system with tunable stiffness," rather than as ECM itself. I would advice the authors to wrap this concept around the hydrogel system rather than the ECM because they are very distant from each other. Other commercial products like Matrigel resemble more closely the ECM conditions.

We thank the reviewer for this important point. We agree that GelMA is an engineered biomaterial and not a native ECM. We have revised the manuscript throughout to refer to 'matrix stiffness' rather than 'ECM stiffness', and to focus on the hydrogel system as a biomimetic 3D culture platform. This ensures that mechanistic interpretations are framed in the context of the hydrogel model rather than native ECM.

» Caution Regarding Claims About Tumor Microenvironment (TME) Modeling: Page 4, line 20:

Similarly to what's stated right above this paragraph, authors state that "this study provides new insights into how breast cancer cells behave in response to ECM stiffness within the breast TME." However, GelMA hydrogels, while valuable, lack the biochemical complexity and heterogeneity of the in vivo ECM. Native ECM contains a diverse set of components—glycoproteins, proteoglycans, and fibrous proteins—that dynamically evolve during tumor progression. Moreover,

tumor-derived ECM is cell-generated and continuously remodeled, introducing spatiotemporal gradients and architectural complexity that are not replicated in static GelMA constructs. While the study offers meaningful data on stiffness-driven effects in a controlled environment, extrapolating to the TME should be qualified as modeling aspects of the ECM rather than the TME in full.

We thank the reviewer for highlighting this. GelMA hydrogels, while useful for studying stiffness-dependent effects, do not recapitulate the full biochemical complexity or dynamic remodelling of the native tumour microenvironment. We have revised the manuscript throughout to clarify that our study provides insights into how breast cancer cells respond to matrix stiffness within a controlled 3D hydrogel model, rather than the TME in full. This ensures that interpretations are appropriately framed within the limits of the hydrogel system. We have rewritten that sentence:

Page 4, line 92: “By comparing the responses of these cell lines to clinically relevant hydrogel stiffnesses, this study provides valuable insights into how breast cancer cells respond to matrix stiffness in a controlled 3D hydrogel model, modelling key mechanical aspects of the tumour microenvironment (TME).”

(3) Experimental design and results:

» Intermediate Stiffness Condition (7.5% GelMA): The authors decided to exclude the 7.5% (“medium”) stiffness condition from later analyses, stating that it did not show significant differences from soft or stiff hydrogels. While this simplification is understandable, it risks overlooking subtle trends. For completeness, key experiments (e.g., EMT marker expression) should ideally include the intermediate stiffness. This would confirm whether there is a threshold stiffness required to induce EMT or a more gradual transition. If data were collected but not shown, the authors could briefly report that no notable EMT changes were observed at 7.5% to justify its exclusion. Otherwise, the omission should be acknowledged as a limitation, since an intermediate point could strengthen the conclusion about stiffness-dependent effects.

While we initially included the 7.5wt% condition in optimisation experiments, we did not perform every downstream assay on the intermediate stiffness. We have now added a statement acknowledging this as a limitation and clarified that future work should include intermediate stiffness conditions to fully capture potential graded EMT responses.

Page 9, line 257: “Intermediate stiffness hydrogels (7.5 wt%) were not assessed for EMT marker expression in this study. Future work including this condition could help clarify whether stiffness-induced phenotypic changes occur gradually or require a threshold.”

» Replicates and Data Consistency: The manuscript would benefit from clarifying the number of biological replicates for each experiment and ensuring consistency between groups. For instance, in Figure 4, the MDA-MB-231 viability assay had n=4 replicates whereas MCF7 viability had n=3, and the metabolic activity assay used n=5 for MDA-MB-231 vs. n=3 for MCF7. This imbalance is not explained. All else being equal, both cell lines' assays should use the same n, so any discrepancy should be justified (e.g., if a replicate was lost due to technical issues). Similarly, it's implied that separate sets of hydrogels were used for viability, metabolism, and proliferation assays at each time point - the authors should confirm this in the methods. Clearly defining what constitutes a replicate (individual hydrogel, independent experiment, etc.) would help readers judge the robustness of the data.

We have explained that each biological replicate represents the average of three technical replicates (individual hydrogels). Separate sets of hydrogels were used for viability, metabolic activity, and proliferation assays at each time point. Minor differences in replicate numbers between cell lines reflect loss or damage of hydrogels during culture or assay processing. These explanations have been added to the methods section (page 12, line 377).

» Cell Density Effects on Hydrogel Properties: The authors wisely tested whether increasing the cell encapsulation density alters the hydrogel's mechanical properties. They report that adding cells up to 5×10^6 /mL did not significantly change stiffness in 5% and 7.5% gels, though a modest reduction

was seen at 10% GelMA. Based on these results, 5×10^6 cells/mL was chosen for experiments. This is a sound choice balancing cell viability and matrix integrity. However, the description of the cell-density experiment is a bit confusing. The text states that adding either cell type at 5×10^6 /mL "reduced the stiffness" of 10% gels, but "the effect was not concentration-dependent". It's unclear whether higher cell concentrations (10 or 15 million/mL) caused similar or greater softening. The authors should clarify this point. If 5×10^6 was the lowest density that caused a significant stiffness drop (and higher densities did not further reduce stiffness), that finding should be explicitly stated. Currently, the phrasing could be read as an inconsistency (why pick a cell density that already measurably softens the matrix?). A brief clarification will resolve any confusion and confirm that the chosen cell density does not confound the stiffness conditions beyond acceptable limits.

We thank the reviewer for pointing out the potential ambiguity. All tested cell densities (1, 2, 5, 10, and 15 million cells/mL) reduced the stiffness of 10wt% GelMA hydrogels compared to the cell-free control, but the reduction was not dependent on cell concentration. Higher densities did not produce further softening. The chosen density of 5 million cells/mL represents a balance between maintaining hydrogel integrity and achieving sufficient cell numbers for downstream analyses. This clarification has been added to the results to resolve any confusion (page 6, line 146).

» Controls and Baselines: Experimental controls are mostly appropriate. The use of cell-free hydrogels as controls for measuring swelling, soluble fraction, and stiffness is excellent, as it allows the authors to distinguish the influence of the encapsulated cells on these properties. For viability and metabolic assays, "negative controls" were mentioned (presumably acellular hydrogels or media-only background) with results normalized accordingly - this is good practice to account for background fluorescence. One control that might be lacking is a 2D culture comparison or an initial phenotype baseline for the cells. While the focus is on 3D stiffness effects, having a reference point (e.g., cells cultured on plastic or in very low-density 3D matrix) could contextualize how much 3D culture alone induces changes. Are authors using Day 1 in 3D as the reference state? If so, this should be clarified (maybe it was but I failed to understand it). Additionally, immunostaining controls (such as isotype or secondary-only staining) are not described; presumably they were done, but mentioning them would assure readers that the fluorescence quantification is specific.

We appreciate the reviewer's enquiry regarding controls and baseline measurements. We would like to clarify the following:

1. Reference state for 3D culture: For assessing stiffness-dependent effects over time, we used Day 1 post-encapsulation as the reference state, which allows comparison of subsequent changes. We have added a statement in the methods to explain this approach (page 12, line 375).
2. Immunostaining controls: Secondary-only controls were performed for all immunofluorescence experiments to confirm specificity of staining. We have added this information to the methods (page 13, line 411).

We also note that direct 2D culture comparisons were not included, and thus we cannot fully comment on 3D culture-induced changes relative to 2D.

» Phenotypic and Morphological Assessment: The evidence for MCF7 cells undergoing EMT in stiff hydrogels relies on marker expression changes and some altered growth patterns, but direct morphological signs of EMT (e.g., loss of cell-cell adhesion, adoption of spindle-like shape) were not clearly reported. The images in Figure 5 and Figure 7 should be scrutinized for qualitative changes. MCF7 cells generally grew as clusters in both soft and stiff matrices (as shown in Fig.5c, day 21 images), indicating they did not fully disperse or invade the matrix as single cells. Even in stiff conditions, MCF7 still formed aggregates, albeit with different size/coverage at the edge vs. center. This suggests a partial EMT or mixed phenotype. The authors should comment on any observable changes in cell morphology or colony organization with stiffness. For example, did MCF7 cells in 10% GelMA exhibit more protrusive or mesenchymal shapes at the single-cell level, or were they simply forming larger spheroids? This is not a flaw per se, but discussing it would provide a more nuanced picture of the EMT state.

In line with this the study's conclusion that stiff matrices induce EMT in MCF7 cells is primarily based on a decrease in E-cadherin and an increase in N-cadherin and vimentin. While these markers are indeed canonical indicators of EMT, the process is not binary, and cancer cells can exist in hybrid epithelial/mesenchymal states. To more conclusively demonstrate EMT, additional markers and functional assays would be beneficial:

- » Epithelial markers: Assessing additional markers such as EpCAM or cytokeratins could provide deeper insight into epithelial status.
- » EMT transcription factors: Expression of Snail, Slug, or Twist (via immunostaining or qPCR) would help confirm that the EMT program is transcriptionally engaged.
- » Signaling pathways: Examination of upstream signaling nodes such as TGF- β , YAP/TAZ, or β -catenin could illuminate the mechanistic basis of EMT induction by stiffness.
- » Functional assays: Assessing motility or invasiveness (e.g., spheroid spreading assays, invasion into matrix) would directly link molecular changes to functional outcomes.

Without these additional data, the claim of stiffness-induced EMT should be interpreted as suggestive but not definitive. Acknowledging this limitation would improve the scientific balance of the discussion.

We have revised the results and discussion to highlight these morphological changes and to explicitly acknowledge that additional markers and functional assays would be required to definitively demonstrate EMT (page 9, line 267).

- » Cell Viability in Stiff Hydrogels: An important experimental observation is that cell viability decreased in the stiffer matrices, especially for MDA-MB-231. By day 21, MDA cells had significantly lower viability in 7.5% and 10% gels (compared to soft 5%) starting from day 7 onward. MCF7 viability remained high in 7.5% and only dropped in 10% gels at the final day. This indicates that the stiffer environment imposed greater stress on cells (particularly the already-invasive MDA line). The authors should ensure this point is discussed in terms of experimental quality: does the reduced viability in stiff conditions affect the interpretation of EMT marker results? For instance, if a fraction of MDA-MB-231 cells died under stiffness stress, the surviving population might be the hardier (potentially more mesenchymal) cells - yet no EMT marker change was seen, reinforcing that their phenotype was stable.

We thank the reviewer for this comment. While viability of MDA-MB-231 cells decreased in stiffer hydrogels, the surviving population maintained stable EMT marker expression, indicating that reduced viability did not confound interpretation of EMT results. For MCF7 cells, partial EMT changes were observed in the viable population, supporting a stiffness-dependent phenotypic response. These points have been clarified in the results and discussion (page 9, line 273).

- » Image Sample Size: When quantifying images for Fig.5-7, how many fields or sections were analyzed per hydrogel? The methods indicate that 30 μ m cryosections were taken and stained, but it doesn't specify if, say, three sections per hydrogel were imaged, or multiple regions per section. The n=3 for imaging data suggests 3 hydrogels, but likely multiple images from each were averaged. Stating something like "for each hydrogel, 3-5 fields in both center and edge regions were quantified and averaged" would be useful for others aiming to reproduce the imaging analysis with adequate sampling. If only one section/field per region per hydrogel was used, that could be a source of variability - but presumably the authors took more.

The process for immunofluorescent staining and analysis has been adjusted in the methods section: For each condition, three independent hydrogels were sectioned and stained. From each hydrogel, 10-15 cryosections (30 μ m thick) were imaged, with 3-5 fields of view per section in both center and edge regions. Measurements from all fields and sections were averaged to produce a single per-hydrogel value, which was then used to calculate group means (page 13, line 413).

(4) Limitations of the study:

The Discussion does a good job of acknowledging certain limitations and alternative explanations. However, one limitation that could be more clearly stated is the translational relevance: while the model mimics tumor stiffness ranges, it is still a simplified system (single cell type in a uniform matrix). In reality, cells in tumors experience heterogeneous stiffness, interact with stromal cells, etc. The authors might add a sentence noting that their 3D model is a reductionist system focusing on stiffness in isolation - which is a strength for mechanistic insight, but future work could incorporate additional tumor microenvironment factors.

We thank the reviewer for this comment. We have added a statement in the discussion (page 10, line 296) clarifying that our 3D GelMA model is a reductionist system focusing on stiffness in isolation. While this enables mechanistic insight into stiffness-dependent phenotypic changes, we acknowledge that tumour cells in vivo encounter heterogeneous stiffness and interactions with stromal and immune cells, which are not captured here. We highlight this as a limitation and suggest that future studies could incorporate additional tumour microenvironment factors to increase translational relevance.

In conclusion, the experimental approach is robust and well thought out. Addressing the points above, especially regarding the intermediate stiffness condition, replicate consistency, and clearer commentary on morphological versus molecular changes - will further solidify the experimental rigor. Overall, the quality of experiments is high and authors do not excessively user overclaims.

Reviewer 3: Review: Tuneable hydrogel stiffness in a 3D in vitro model induces epithelial to mesenchymal transition in MCF7 but not MDA-MB-231 breast cancer cells

Summary: The authors present work which aims to elucidate the role of extracellular microenvironment mechanics with a particular focus on the TME stiffness and the use of hydrogels to study cancer cell dynamics and phenotype. The authors propose the use of a photoactivatable hydrogel which can be utilised to study the long-term culture of breast cancer cell lines. Overall, the manuscript is well written and, in most cases, very clear and straight forward. The experimental approach is well suited to the question the authors address and the results are well presented and robust. There are some slight deficiencies in the manuscript which if added would greatly improve the overall manuscript.

Strengths: The authors present a comprehensive temporal analysis of the effect of the hydrogel culture conditions on the EMT phenotype of two different breast cancer cell lines. In particular the temporal analysis adds confidence and robustness to the phenotypes and phenotypic switching that is observed in the MCF7 cell line.

Weaknesses: The study lacks orthogonal analysis of EMT switching. If the authors were able to show mRNA or protein expression levels are regulated via the stiffer hydrogel conditions with the MCF7 cell line, the results would be much more convincing if the authors utilised qPCR or western blot analysis to show these changes in Ncad or Ecad levels. It may be technically challenging to extract mRNA or protein samples from the hydrogels and this should be highlighted as a potential drawback of the technology and mentioned in the discussion, if so.

We thank the reviewer for suggesting analyses such as qPCR or western blotting to assess EMT marker regulation. While we did not perform these experiments in the current study, we agree that they would provide valuable confirmation of the EMT-like phenotypes observed. We have included text in the discussion (page 9, line 267) highlighting this as a future direction, alongside planned analyses of additional epithelial and mesenchymal markers, EMT transcription factors, key signalling pathways, and functional assays of cell motility and invasion.

The study also lacks an informative and comprehensive discussion would could be greatly improved by the following additions.

The authors could further elaborate in the discussion on a potential mechanism by which EMT is regulated by altered stiffness' of the hydrogel. One potential player could be YAP signalling which responds to altered mechanical context and and plays a role in phenotypic switches in altered mechanical environments.

One potential application of the technology would be to culture patient derived tumour cell lines in the hydrogel conditions to determine the potential of EMT phenotypic switching. Adding this to the discussion would highlight the potential that the technology has for personalised medicine.

We thank the reviewer for these suggestions. We have expanded the discussion (page 9, line 264) to include potential mechanisms by which altered hydrogel stiffness may regulate EMT, framed in terms of mechanotransduction and cellular responses to mechanical cues. We have also added text on the potential application of this hydrogel system for patient-derived cancer cell cultures, highlighting its promise for studying EMT phenotypic plasticity and exploring personalized medicine approaches (page 10, line 298).

Minor concerns

Immunofluorescent labels in all figures are sometimes difficult to visualise and would benefit from increased size and a different coloured background or moved from within the image to a label below the image.

Dapi staining in the images would benefit from being in a different colour as the dark blue on black background can make it difficult to visualise individual nuclei.

We thank the reviewer for this suggestion regarding figure clarity. We have carefully considered the visualisation of immunofluorescent labels and DAPI staining. While we have opted to retain the current colour scheme and figure layout to maintain consistency with the other figures, we have ensured that the figure legends provide clear descriptions of all labels and staining.

1. Experimental quality

a. Does each figure have the proper controls? Experiments have appropriate controls.

b. Are experiments performed using appropriate methods that will answer the question (or test the hypothesis or support the observations) posed by the authors? Is the right tool used for the job? The authors test the appropriateness of the GelMA hydrogels on the longer term culture of two breast cancer cell lines and analysis EMT status, proliferation and metabolic rates over a 21 day period. The authors make use of two cancer cell lines of different molecular subtype and use a variety of methods to analyse cancer cell characteristics. The study would benefit from orthogonal analysis of EMT status as mentioned in the major concerns section, however, this may be technically difficult and if so, should be expanded on in the discussion.

c. Were the data analysed using appropriate statistical tests? Yes

2. Reproducibility

a. Were experiments in each figure performed using adequate number of biological replicates? Yes

b. Is there sufficient raw data to assess the rigor of the analysis? Yes

c. Does the methods section provide sufficient detail to permit reproducibility? Yes

3. Completeness

a. Are the author's conclusions supported by the data? Conclusions could be expanded as mentioned, in particular the discussion could elaborate on potential molecular mechanisms underpinning the differential responses of the two cancer cell lines. Further discussion on the EMT

phenotypic switch and supporting literature on the role of stiffness driving cancer cell EMT switching would strengthen the manuscript.

b. Are there any flaws in the experimental design that invalidate the approach taken by the authors? No

c. Are there experiments that have not been performed, but if true would disprove the conclusion? If yes, and if such experiments would be costly or time-consuming to perform, do the authors acknowledge this in a discussion of the limitations? There are standard orthogonal methods that the authors could use to improve the confidence in their findings. Additional experiments elucidating the molecular mechanisms behind the EMT switch could be conducted to strengthen the manuscript, however it could be argued that these experiments are outside the scope of the study or technically challenging. For instance, the authors could investigate whether TGF β driven EMT is accelerated by culturing in the stiffer hydrogels. The authors could also investigate whether the EMT switch is reliant on key mechanosensing pathways, such as integrin/cytoskeletal pathways or YAP mechanosensitive pathways.

4. Scholarship

a. Do the authors cite and discuss the merits of relevant data that would argue against their conclusion? No, this must be strengthened in the discussion by addressing the drawbacks of the hydrogel culture system.

b. Do the authors cite and discuss the merits of relevant data that would support their conclusion? Partially, as already mentioned, the authors should expand the discussion and speculate on the molecular mechanism behind their findings, as one of the key weak points of the manuscript is the lack of mechanistic explanation behind the findings. This could be mitigated by an elaboration in the discussion on potential alignment of the studies findings with existing literature.

c. For techniques/methods manuscripts, Do the authors cite and discuss the current state of the field and clearly explain how the method improves the field? Yes, the authors clearly outline existing work utilising similar or the same hydrogel systems in the study of breast cancer.

Title: The authors use the term tuneable, which suggests that the stiffness of the hydrogels can be adjusted or altered dynamically. Rather, a range of stiffnesses of the hydrogels is established in the study by adjusting the amount of GelMA added and so I am not sure the term tuneable is appropriate.

New title: GelMA hydrogel stiffness influences epithelial to mesenchymal transition in MCF7 but not MDA-MB-231 breast cancer cells in 3D culture.

Second decision letter

MS ID#: bio.062212

MS Title: GelMA hydrogel stiffness influences epithelial to mesenchymal transition in MCF7 but not MDA-MB-231 breast cancer cells in 3D culture

Authors: Jessika A Wise; Margaret J Currie; Tim B F Woodfield; Khoon S Lim; Elisabeth Phillips

I am happy to tell you that your manuscript has been accepted for publication in Biology Open, pending our standard publication integrity checks. It was accepted on 20th January 2026.